# An Investigation of Real-Time Galileo/GPS Integrated Precise Kinematic Time Transfer Based on Galileo HAS Service

**DOI:** 10.3390/s25103243

**Published:** 2025-05-21

**Authors:** Lei Xu, Shaoxin Chen, Yuanyuan An, Pengli Shen, Xia Xiao, Qianqian Chen, Jianxiong Wei, Yao Chen, Ye Yu

**Affiliations:** 1Nanjing Institute of Metrological Supervision and Testing, Nanjing 210049, China; xu495587@126.com (L.X.); chenqianqian_js@petalmail.com (Q.C.); wjxhero@126.com (J.W.); chy519@petalmail.com (Y.C.); 2State Key Laboratory of Estuarine and Coastal Research, School of Marine Sciences, East China Normal University, Shanghai 200062, China; 3China Satellite Network System Co., Ltd., Beijing 100020, China; anyuanyuantianwen@163.com; 4National Time Service Center, Chinese Academy of Sciences, Xi’an 710600, China; xiaoxia@ntsc.ac.cn; 5University of Chinese Academy of Sciences, Beijing 100049, China; 6Key Laboratory of Time Reference and Applications, Chinese Academy of Sciences, Xi’an 710600, China; 7Rocket Force University of Engineering, Xi’an 710025, China; yuye115@mails.ucas.edu.cn

**Keywords:** Galileo HAS, time transfer, PPP kinematic, GPS, Galileo

## Abstract

GNSS Precise Point Positioning (PPP) technology has been extensively applied to post-processing international comparisons between UTC/TAI times and real-time time transfer, predominantly in static configurations. However, with the swift advancement of intelligent and unmanned systems, there is an urgent need for research into kinematic time transfer. This paper introduces a kinematic model Galileo/GPS integrated PPP time transfer approach leveraging the Galileo High Accuracy Service (HAS). The study utilized observational data from seven stations spanning 22 days. The findings indicate that under static conditions, GPS, Galileo, and Galileo/GPS PPP, when supported by the Galileo HAS, can achieve time transfer with sub-nanosecond precision. In kinematic scenarios, the accuracy of single-system PPP time transfer is comparatively lower, with frequent re-convergence events leading to significant accuracy degradation (exceeding 1 ns). However, in cases where re-convergence is infrequent due to a limited number of satellites, sub-nanosecond time transfer is still attainable. The Galileo/GPS integrated PPP time transfer effectively mitigates the issue of re-convergence, ensuring sub-nanosecond accuracy across all links (0.48 ns). Consequently, it is recommended to employ a multi-system integration approach for kinematic PPP time transfer, particularly when utilizing the Galileo HAS. In terms of frequency stability, GPS, Galileo, and Galileo/GPS PPP demonstrate short-term stability (over 960 s) of (5.29 × 10^−13^, 3.34 × 10^−13^, and 1.60 × 10^−13^), respectively, and long-term stability (over 15,360 s) of (1.49 × 10^−13^, 1.02 × 10^−13^, and 4.06 × 10^−14^), respectively.

## 1. Introduction

Time information plays a crucial role in many industries, such as power systems, financial systems, intelligent transportation, and military applications. There are currently various methods for time transfer, such as satellite two-way satellite time and frequency comparison technology (TWSTFT), fiber-optic time transfer technology, and GNSS technology [1,2]. The TWSTFT method cannot be widely applied due to its expensive terminals. In addition, the application range of fiber optic time synchronization is also limited due to the high cost issue. Compared to the aforementioned methods, GNSS has its own unique advantages. The cost of time transfer using GNSS is lower, and it is more convenient to use GNSS for time transfer. More importantly, the current GNSS technology has been officially adopted by the BIPM for international UTC (Coordinated Universal Time) time comparisons [3,4,5].

As satellite navigation systems advance at a breakneck pace, time transfer applications based on GNSS satellites have become one of the widely used timing methods [6,7]. The original GNSS time transfer technology used the common view (CV) time transfer technique, but because the CV time transfer technique requires simultaneous observation of the same satellite, it is limited by distance [8]. With the provision of precise satellite orbit and clock offset products by the IGS (International GNSS Service), scholars have further developed the all-in-view (AV) time transfer technology. This technology is not limited by distance and can achieve the same level of time transfer accuracy as CV [9]. Nevertheless, the aforementioned time transfer methodologies are constrained by the precision of GNSS pseudorange measurements. Given that the accuracy of GNSS carrier phase observations surpasses that of pseudorange observations by a factor of two, researchers have subsequently refined the technology for precise point positioning (PPP) time transfer [5,6,7,10]. Within the recent past, the research domain of PPP time transfer has experienced substantial growth and refinement. The evolution has been notable, moving from the confines of single-system PPP time transfer to the broader capabilities of multi-system synchronization [11,12,13]. Additionally, it has advanced from the limitations of single-frequency measurements to the more nuanced accuracy afforded by multi-frequency PPP time transfer [14,15,16]. Most notably, the shift from the float solution to the fixed solution in PPP time transfer [17,18] has marked a significant leap in precision and reliability. The research has evolved from estimating receiver clock errors using white noise estimation to parameter estimation using a clock model for receiver clock errors [6,19,20]. Yet, the studies on PPP time transfer previously mentioned have largely been centered around international UTC evaluations and post-processing time transfer endeavors. With the rapid development of society, users’ demands for real-time performance are increasing, making the research on real-time PPP time transfer an urgent priority. Thankfully, the landscape of real-time PPP time transfer has seen significant progress since 2013, with the International GNSS Service (IGS) making real-time data streams readily available. This development has catalyzed scholarly exploration into the capabilities of real-time PPP, leading to groundbreaking achievements in accuracy. Researchers have now mastered the art of real-time PPP time transfer, attaining a level of precision that is astonishingly within the sub-nanosecond realm [21]. Subsequently, based on the real-time PPP time transfer method, scholars have calculated the real-time clock differences to UTC(k) time or GNSS time and broadcasted them to users. After receiving these real-time satellite orbit and clock difference products, users can utilize the PPP algorithm to achieve one-way precise time synchronization [22,23]. Their results indicate that real-time one-way PPP timing can achieve timing accuracy better than 0.5 nanoseconds. Ge et al. [24] have also achieved PPP timing using the IGS real-time stream. Their method is not limited by the real-time precise orbit products of any analysis center and is applicable to various IGS real-time streams, with an accuracy that can reach the sub-nanosecond level. Moreover, the reliability of real-time products is often contingent upon the stability of internet connections, which, if erratic, can readily trigger the re-convergence of receiver clock parameters. This susceptibility underscores the need for robust and consistent data transmission to ensure the integrity of real-time PPP applications. Ge et al. [19] have further studied that the receiver clock offset model can further enhance real-time PPP time transfer, reduce the frequency of re-convergence, and at the same time reduce the noise of receiver clock offset, thereby improving the accuracy of real-time PPP time transfer.

Nevertheless, the transmission of correction data via the internet continues to pose challenges to the reliability of PPP time transfer. Fortunately, the current BeiDou-3 satellites provide a precise point positioning service (PPP-B2b) through the B2b signal in the Asia-Pacific region [25,26], and Galileo also offers a high-precision service (HAS) globally [27]; both of these services directly broadcast correction data to users through satellite signals. HAS is the European free PPP service provided by Galileo. On 24 January 2023, the European Commission announced the start of its initial services. The message provides orbit, clock, code biases, and soon phase biases for Galileo E1-E5a-E5b-E6 and GPS L1C/A-L2C signals. The message corrects the Galileo I/NAV and GPS LNAV L1 C/A combinations and is transmitted in the Galileo E6 signal (E6-B data component) and through the Galileo HAS internet data distribution (IDD). With the aforementioned two services, users can directly achieve PPP time transfer without the internet, and the reliability is better. Ge et al. [7] have evaluated and analyzed real-time BDS-3 PPP time transfer via the PPP-B2b service, achieving a real-time PPP time transfer accuracy of 0.3 ns. Zhang et al. [27] and Mao, et al. [28] have conducted a preliminary analysis of Galileo PPP time transfer through the Galileo HAS and have achieved a time transfer accuracy of 0.5 ns, although scholars have achieved BDS-3 or Galileo PPP time transfer without a network environment. However, current research mainly focuses on PPP time transfer or timing in static models. With the rapid development of unmanned systems and intelligent driving, real-time kinematic high-precision timing methods have become an extremely urgent need. To this end, this paper intends to establish a real-time Galileo/GPS integrated PPP time transfer in a kinematic model using the Galileo HAS service to meet the high-precision timing needs of current intelligent systems or unmanned systems.

The layout of this paper is outlined as follows: The second section presents the fundamental theory behind real-time PPP time transfer in a kinematic model. In the third section, the strategies used in the experiment and the data used are introduced. In the fourth part, six time links are used to evaluate and verify the real-time PPP time transfer in a kinematic model pointed out in this paper. Finally, some conclusions and discussions are given.

## 2. Materials and Methods

We outline the methodology for PPP time transfer. Subsequently, we offer an in-depth discussion on the retrieval of accurate satellite orbits and clock data using broadcast ephemerides along with HAS corrections.

### 2.1. PPP Time Transfer Model

We deploy a dual-frequency ionosphere-free (IF)combination within the PPP time transfer model [29]. Leveraging precise satellite products and mitigating various error sources, the pseudorange and carrier phase observation equations are formulated as follows:(1)$$p_{r,IF}^{s}=h_{r}^{s}\cdot \Delta x+cdt_{r,IF}+m_{r,w}^{s}\cdot Z_{w}+\xi_{r,\text{IF}}^{s}$$(2)$$l_{r,\text{IF}}^{s}=h_{r}^{s}\cdot \Delta x+cdt_{r,\text{IF}}+m_{r,w}^{s}\cdot Z_{w}+N_{r,\text{IF}}^{s}+\psi_{r,\text{IF}}^{s}$$(3)$$\left\{\begin{array}{l} cdt_{r,\text{IF}}=cdt_{r}+cd_{r,\text{IF}}\\ d_{r,\text{IF}}=\alpha_{ij}d_{r,i}+\beta_{ij}d_{r,j}\\ \alpha_{ij}=\frac{f_{i}^{2}}{f_{i}^{2}-f_{j}^{2}};\beta_{ij}=-\frac{f_{j}^{2}}{f_{i}^{2}-f_{j}^{2}};i\neq j \end{array}\right.$$

Here, $l_{r,\text{IF}}^{s}$ and $p_{r,\text{IF}}^{s}$ represent the OMC (Observed Minus Computed) values for IF carrier phase and code observations, respectively. The subscripts *s* and *r* denote the satellite and receiver, while Δx signifies the position increment.

Here, $h_{r}^{s}$ represents the coefficients for the position. The indices *i* or *j* differentiate between the different frequencies. $\alpha_{ij}$ and $\beta_{ij}$ is the frequency factor. For Galileo and GPS, the E1/E5a and L1/L2 frequency are chosen. $d_{r,i}$ denotes the hardware delay; $f_{i}$ is the frequency *i*; $dt_{r,IF_{ij}}$ displays the combined receiver clock offset in the IF combination, inclusive of the hardware delay; $dt_{r}$ is the receiver clock offset; $d_{r,IF}$ indicates the hardware delay for the IF combination; $Z_{w}$ is the zenith wet delay; $N_{r,IF}^{s}$ is the float IF ambiguity; $m_{r,w}^{s}$ is the wet mapping function; and $\xi_{r,IF}^{s}$ or $\psi_{r,IF}^{s}$ are the noise terms.

In addition, based on elevation-dependent weighting used in our work, the precision of orbit and clock calculated from the HAS correction also need to be considered. The weight W can be expressed as [5](4)$$\left\{\begin{array}{l} W=diag(\sigma_{1}^{-2},\sigma_{2}^{-2},\sigma_{3}^{-2},\cdots ,\sigma_{m}^{-2})\\ \sigma^{2}=FR_{r}(a_{\sigma }^{2}+b_{\sigma }^{2}/\sin E^{2})+\sigma_{eph}^{2} \end{array}\right.$$
where F is the satellite system error factor; $R_{r}$ indicates code/carrier phase error ratio; $a_{\sigma }$ and $b_{\sigma }^{2}$ are carrier-phase error factors a and b (m); E illustrates elevation angle; $\sigma_{eph}$ is the URA from the HAS service; m is the number of satellites.

Distinct from the Multi-GNSS experiment (MGEX) final products and the real-time product offerings by IGS, the Galileo HAS service adopts Galileo System Time (GST) as the benchmark for clock offsets [27], thereby achieving improved stability and continuity. However, the GPS satellite clock references are not synchronized to a common time standard; this discrepancy does not impact the accuracy of the timing process. The receiver clock offset $cdt_{r,IF_{ij}}$, as derived from the PPP model utilizing Galileo data, encompasses the discrepancy between Galileo System Time (GST) and the local time reference, as well as the inherent hardware delay. Typically, such hardware delays are pre-calibrated to ensure accuracy [30].

The divergence D between the time *T* and GST can be articulated as follows:(5)D=T−GST

Here, D signifies the local time of the user’s receiver. For the PPP one-way timing, Equation (5) can be used directly.

For two stations, the divergence ΔT will be written as(6)$$\begin{array}{ll} \Delta T & =D_{1}-D_{2}\\ & =T_{1}-\mathrm{GST}-(T_{2}-\mathrm{GST})\\ & =T_{1}-T_{2} \end{array}$$

Here, *T*_1_ and *T*_2_ indicate the time of users. Due to the lack of a reference in one-way PPP timing, which makes it impossible to verify the timing accuracy of kinematic PPP, we used time transfer results to validate the performance of kinematic PPP time transfer. During the PPP time transfer process, data loss may occur due to data exchange. This paper focuses on studying the kinematic PPP time transfer performance under conditions where no data are lost, while considering potential future data loss scenarios; future work could incorporate a receiver clock offset prediction model.

### 2.2. Real-Time Recovery of Precise Orbit and Clock Correction Products

The precise orbit can be constituted by the combination of broadcast ephemeris and Galileo HAS corrections [25,31], which is articulated as:(7)$$\partial COO(t)=\left[\begin{array}{c} \partial COO_{r}\\ \partial COO_{a}\\ \partial COO_{c} \end{array}\right]+\left[\begin{array}{c} \delta_{r}\\ \delta_{a}\\ \delta_{c} \end{array}\right](t-t_{0})$$

Here, ∂COO(t) is the orbit correction parameters *t*, where $\partial COO_{r}$, $\partial COO_{a}$, $\partial COO_{c}$ denotes the corrections in the RAC (Radio, Along, Cross) directions. $\delta_{r}$, $\delta_{a}$, $\delta_{c}$ represent the rates of change in the RAC directions. Subsequently, the calculation of the rotation matrix components for each direction is required.(8)$$\begin{array}{l} e_{a}=\frac{r}{\left|r\right|}e_{c}=\frac{r*(\dot{r})}{\left|r*(\dot{r})\right|}e_{r}=e_{a}*e_{c}\\ \partial x(t)=[e_{r}e_{a}e_{c}]\partial COO(t) \end{array}$$

Here, ∂x(t) is the HAS correction parameters. $e_{r},e_{a},e_{c}$ are the coefficients in the RAC directions, respectively. $\dot{r}$, r denote the velocity and position vectors of the satellite.(9)∂pcoo(t)=∂bcoo(t)+∂x(t)

Here, ∂bcoo(t) denotes the broadcast ephemeris coordinate. ∂pcoo(t) is the precise coordinates.

The precise clock correction values are needed.(10)∂pclk(t)=∂bclk(t)+∂CLK(t)/c

Here, ∂pclk(t) is the precise clock, ∂bclk(t) is the broadcast ephemeris clock, and *c* stands for the speed of light.

## 3. Experimental Data and Strategy Selection

### 3.1. Dataset

Seven GNSS stations, each equipped with an H-master clock, were selected for this study and are listed in Table 1. Their geographical distribution is depicted in Figure 1. The PTBB station serves as the central node. The observational data were collected over a 22-day period, ranging from DOY (Day of year) 165 to 187 in 2023. The precise orbit and clock products will be derived by integrating the broadcast ephemeris with the Galileo HAS corrections. The IGS final products were obtained by downloading from the ftp server at ftp://igs.gnsswhu.cn/ (from DOY 165 to187 in 2023, accessed on 13 April 2025). The GPS PPP time transfer solutions, derived from IGS final products using a static model, are established as the benchmark reference.

### 3.2. Processing Strategy

In our research, the approaches for PPP time transfer are succinctly detailed in Table 2. We have incorporated and corrected known errors, such as relativistic effects and the Sagnac effect, by applying the IERS standard models [29]. The receiver clock offset is modeled as white noise. We have developed PPP time transfer strategies for Galileo, GPS, and their integrated Galileo/GPS systems, leveraging the Galileo HAS service. The time transfer from the IGS final clock products serve as our reference standard. To ensure the convergence of the PPP, we excluded the initial day’s data from our precision calculations. In the Galileo/GPS PPP model, an ISB parameter is included. The receiver clock offset is set relative to Galileo observations, and for GPS, it comprises both the Galileo receiver clock offset and the ISB. Notably, in static PPP time transfer, the receiver’s coordinates are treated as a constant, while in kinematic PPP time transfer, they are regarded as white noise. In addition, the Melbourne–Wubbena (MW) and geometry-free (GF) combination are used for cycle slip detection [32].

## 4. Results

This section begins by presenting the performance of single-constellation Galileo, single-constellation GPS, and combined Galileo/GPS PPP time transfer in static mode. Subsequently, it investigates and analyzes the performance of single-constellation Galileo, single-constellation GPS, and combined Galileo/GPS PPP time transfer in kinematic mode.

### 4.1. PPP Time Transfer in Static Model with HAS Service

Figure 2 illustrates the receiver clock offsets at the PTBB station, calculated using GPS PPP time transfer in a static model, and Figure 3 provides a comparative analysis of the receiver clock offsets for six additional stations under static conditions. Synthesizing the information from both figures, we can discern several key takeaways. Notably, a striking consistency in the trends of receiver clock offsets across various stations is apparent when utilizing GPS PPP in conjunction with the Galileo HAS service. This uniformity suggests that the receiver clock offsets, as determined by PPP, largely reflect the discrepancy between the local time at each station and the reference time indicated by the HAS service. Given that the stations under study are primarily from timekeeping laboratories, where the local time is carefully regulated, the observed trends in receiver clock offsets are predominantly shaped by the reference standards inherent in the HAS service products. Furthermore, it was observed that although the receiver clock offsets at various stations exhibit a pronounced similarity in their trends, there are noticeable systematic variations among them. These variations are largely attributed to the inconsistencies in hardware delays across different receivers. As the calibration of hardware delays is a complex process and does not constitute the main focus of this paper, and considering that hardware delays tend to be stable over short durations, their impact on the research presented here is deemed negligible [30]. Thirdly, a close examination of Figure 2, especially the enlarged detail, reveals that the GPS PPP calculations relying on the Galileo HAS service exhibit minor yet discernible fluctuations in the receiver clock offsets. While these fluctuations are not significant, they can impact the precision of one-way timing derived from GPS PPP. Consequently, it can be deduced that GPS PPP reliant on the HAS service may not be adequate for applications requiring high-precision one-way timing. The primary reason behind this is the absence of a standard time reference for the GPS clock offsets in the Galileo HAS service [27], a characteristic it shares with the BDS-3 PPP-B2b [7].

Figure 4 illustrates the time transfer outcomes for the six time chains designed with the PTBB station as the central node. Several conclusions can be inferred from this figure. Firstly, the time transfer results have resolved the apparent discontinuities present in Figure 2, thereby corroborating that the erratic fluctuations in the receiver clock offsets observed earlier were attributable to the GPS reference discrepancies within the HAS service data. Secondly, there is a noticeable inconsistency in the time transfer results across different chains, suggesting that even though all the stations selected are synchronized with UTC(k), each station exhibits unique characteristics. Thirdly, occasional jumps in the time transfer results, such as those seen in the NIST-PTBB link, are primarily due to a paucity of available satellites at certain times, necessitating a reinitialization of the PPP calculations. Lastly, it is evident that the time transfer results for the USN7-PTBB and USN8-PTBB chains are nearly identical, which, beyond differing hardware delays, can be mainly attributed to the fact that the USN7 and USN8 stations are both connected to the same atomic clock source.

In order to delve deeper into the performance of single-constellation Galileo and combined Galileo/GPS PPP time transfer in static mode, Figure 5 depicts the receiver clock offsets at the PTBB station derived from single-constellation Galileo and Galileo/GPS combined PPP solutions in static mode. Furthermore, Figure 6 supplements this analysis by displaying the receiver clock offsets for six other stations, also determined under static mode conditions using single-constellation Galileo and Galileo/GPS integrated PPP calculations.

Upon comprehensive analysis, a few key insights emerge. First and foremost, there is a remarkable consistency between the receiver clock offsets determined by Galileo-only and Galileo/GPS integrated PPP, which can be largely attributed to the prioritization of receiver clock offsets from Galileo observation equations in the integrated Galileo/GPS PPP process. Secondly, it is noted that the Galileo-only PPP calculations experience several instances of re-convergence, likely due to the scarcity of available satellites. Conversely, the receiver clock offsets from the Galileo/GPS combined PPP exhibit a more consistent and stable time series, devoid of any re-convergence events. A closer examination of Figure 7, which details the satellite count utilized in the PPP calculations for GPS-only, Galileo-only, and Galileo/GPS combined PPP, further substantiates this, indicating that at certain intervals, the Galileo-only approach might encounter situations where the number of satellites drops below the threshold of four. Lastly, a comparison between Figure 2 and Figure 5 reveals that the receiver clock offsets from Galileo and Galileo/GPS PPP calculations are not only free from abrupt jumps but also remarkably stable. This stability can be primarily attributed to the fact that the Galileo satellite clock offsets in the HAS service are aligned with the GST, whereas the GPS satellite clock offsets are not subjected to the same standardization. Therefore, it can be concluded that Galileo-only or Galileo/GPS PPP, facilitated by the Galileo HAS service, is apt for one-way time synchronization applications.

Figure 8 presents the outcomes of the time transfer assessments conducted in static mode, utilizing Galileo-only and the integrated Galileo/GPS PPP approaches. A comparison with Figure 4 reveals that the time transfer results from either Galileo-only or the Galileo/GPS combined PPP are in alignment with the general trends observed in the GPS-only PPP time transfer. This concurrence is logically sound, as the essence of PPP time transfer is to serve as an intermediary, with the ultimate objective of ascertaining the temporal disparity between two distinct locations, irrespective of the specific satellite constellation involved. Furthermore, mirroring the earlier observations on receiver clock offset behavior, the time transfer results for Galileo PPP exhibit multiple instances of re-convergence, an issue that is prevalent in scenarios where the satellite coverage or geometry is suboptimal. In contrast, the Galileo/GPS combined PPP time transfer demonstrates a consistent and stable performance, effectively circumventing the re-convergence phenomenon. This affirms that the integration of multiple GNSS systems is highly beneficial, if not essential, for achieving precise and reliable time synchronization across the globe, particularly when leveraging the Galileo HAS service. The fusion of Galileo with GPS enhances the satellite availability and geometric distribution, thereby providing a more robust framework for high-precision timing applications.

In order to conduct a thorough numerical analysis of the performance of GPS, Galileo, and the integrated Galileo/GPS PPP time transfer in static mode, we have utilized the GPS PPP time transfer results derived from IGS final products as a benchmark. We have computed the standard deviation (STD) of the discrepancies between the time transfer outcomes for GPS, Galileo, and the combined Galileo/GPS PPP when compared to their counterparts based on IGS final products, with the results summarized in Table 3. Furthermore, Figure 9 illustrates the modified Allan variance deviation (MDEV) values for the time transfer performance of GPS, Galileo, and Galileo/GPS PPP across six distinct time links. This comprehensive analysis provides a detailed assessment of the temporal stability and precision of the various PPP time transfer methodologies under static conditions. Upon a meticulous review of Table 3 and Figure 9, we can distill several key insights. Firstly, the data from Table 3 clearly indicate that the GPS, Galileo, or Galileo/GPS integrated PPP time transfer, facilitated by the Galileo HAS service in static model, is capable of achieving timing accuracy at the sub-nanosecond level (specifically, better than 0.5 ns). Secondly, a closer look reveals that the Galileo PPP time transfer exhibits slightly better precision (with an average STD value of 0.31 ns) compared to the GPS PPP time transfer (with an average STD value of 0.41 ns). This marginal improvement is likely the result of two primary factors: the use of hydrogen masers onboard Galileo satellites, which are known for their stability; and the superior accuracy of the Galileo satellite clock offset products provided by the Galileo HAS service when compared to those for GPS satellites [27]. Thirdly, the combined Galileo/GPS PPP time transfer (with an average STD value of 0.30 ns) demonstrates superior performance to the Galileo-only PPP time transfer. This enhancement in performance is mainly attributed to the increased reliability that comes with the fusion of two GNSS systems, effectively circumventing the issue of re-convergence that can be triggered by an insufficient number of satellites. This pattern is also discernible in the aforementioned figures, further validating our observations.

When it comes to frequency stability (Table 4), the performance of GPS, Galileo, and the integrated Galileo/GPS PPP time transfer, all augmented by the Galileo HAS service, mirrors the previously discussed outcomes. The GPS PPP time transfer, as indicated by its modified Allan deviation (MDEV), performs the least effectively, with Galileo PPP exhibiting slightly better stability. However, the combined Galileo/GPS PPP time transfer stands out with the most impressive results. This indicates that the synergistic use of Galileo and GPS systems provides more robust frequency stability in the context of PPP time transfer. Regarding the frequency stability at the 960 s mark, the average frequency stabilities for GPS, Galileo, and the Galileo/GPS combined PPP in static mode are recorded as (5.80 × 10^−14^, 4.82 × 10^−14^, 4.07 × 10^−14^), respectively. When considering the frequency stability at the 15,360 s interval, the average frequency stabilities for GPS, Galileo, and the Galileo/GPS combined PPP in static mode are determined to be (1.19 × 10^−14^, 1.13 × 10^−14^, 8.69 × 10^−15^), respectively. These figures underscore the superior performance of the Galileo/GPS fusion approach in maintaining stability over extended periods, which is a crucial attribute for high-precision time transfer applications using Galileo HAS service.

### 4.2. PPP Time Transfer in the Kinematic Model Using HAS Service

To further explore the performance of kinematic PPP time transfer based on the Galileo HAS service, Figure 10 presents the receiver clock offsets for the PTBB station calculated using GPS PPP in dynamic mode, while Figure 11 illustrates the receiver clock offsets for six other stations derived from GPS PPP in the kinematic model. Upon examining the data presented in the two figures, a few key takeaways emerge. Firstly, akin to the static scenario, the receiver clock offsets across all stations maintain a consistent trend regardless of whether they are determined by static or kinematic PPP, a pattern largely attributable to the benchmarks of the GPS satellite clock products within the HAS service. Secondly, it becomes evident that the receiver clock offsets derived from kinematic PPP exhibit greater noise compared to those from static PPP. This increase in noise is likely due to the pronounced correlation between the receiver clock offset parameters and the vertical coordinate during the dynamic PPP solution process, a relationship well-documented in the scholarly literature. Lastly, it is observed that the kinematic PPP-derived receiver clock offsets undergo multiple instances of frequent re-convergence, a phenomenon not observed in the static PPP calculations. This susceptibility to re-convergence in kinematic PPP is primarily a consequence of the challenges posed by an insufficient number of satellites during the solution process, leading to more frequent re-initializations of the clock offset estimates. Solving this problem might be possible through the establishment of a receiver clock offset model [30], which will also be part of our future research work.

In our quest to delve deeper into the receiver clock offsets, we conducted an in-depth analysis focusing on the Galileo and Galileo/GPS integrated PPP kinematic model. Figure 12 and Figure 13 present a comparative analysis of the receiver clock offsets, with Figure 12 highlighting the data from the PTBB station and Figure 13 extending this analysis to encompass six additional stations. This visual representation of data provides a comprehensive overview of the performance characteristics of the PPP kinematic model across diverse stations, offering valuable insights into the behavior of receiver clock offsets in kinematic conditions. Mirroring the preceding results, it is observed that the kinematic scenario notably amplifies the noise associated with receiver clock offsets. Moreover, the Galileo-only PPP, which relies on the Galileo HAS service, experiences quite frequent instances of re-convergence in its receiver clock offsets when operating in the kinematic mode, a phenomenon particularly pronounced at the TWTF station. The underlying causes for this behavior have been elucidated in the earlier sections and will not be reiterated here. What is particularly exciting is that the Galileo/GPS combined PPP solution demonstrates the ability to maintain a smooth and continuous time series under kinematic conditions, devoid of any significant re-convergence events. This robust performance can be attributed to the enhanced spatial configuration and increased satellite count that results from the integration of multiple GNSS systems. From these observations, we can confidently conclude that for the implementation of PPP time transfer in kinematic scenarios facilitated by the Galileo HAS service, the combined Galileo and GPS PPP solution is the recommended approach. This recommendation is based on its superior performance in maintaining precision and continuity in time transfer, which is crucial for applications requiring high-precision timing information during movement.

Figure 14 and Figure 15 depict the outcomes of the GPS PPP time transfer, facilitated by the Galileo HAS service, in kinematic mode across six time links, as well as the time transfer results for Galileo and the integrated Galileo/GPS PPP in kinematic mode across the same six time links. Upon examining the data presented in the two figures, several noteworthy observations emerge. Firstly, it becomes clear that the time transfer outcomes still exhibit instances of re-convergence, indicating that this phenomenon is not entirely mitigated. Secondly, aside from the TWTF-PTBB time link, the time transfer noise under the Galileo PPP kinematic mode is observed to be less pronounced compared to the GPS PPP across the remaining links, suggesting a certain level of superiority in noise management. Lastly, while the Galileo/GPS combined PPP demonstrates the most favorable performance in kinematic conditions, there is a noticeable increase in time transfer noise when compared to the static model, highlighting a persistent challenge in the kinematic environments.

To assess the performance of PPP time transfer under kinematic conditions enabled by the Galileo HAS, Table 5 details the standard deviations of the deviations from the reference values for GPS, Galileo, and the combined Galileo/GPS PPP time transfer in dynamic scenarios. Furthermore, Figure 16 depicts the MDEV values for six time links, showcasing their stability in the kinematic PPP operational model. Upon examining the data presented, it is evident that the precision of single-constellation PPP time transfer, either using GPS or Galileo, in kinematic conditions is adversely affected by the amplification of noise and the challenges associated with re-convergence, resulting in accuracies that can exceed 1 nanosecond. This degradation in precision is largely attributed to the frequent re-convergence events. However, it is noteworthy that time links with less frequent re-convergence instances, such as the BRUX-PTBB link, continue to deliver sub-nanosecond time transfer performance in the kinematic model. Fortunately, the combined Galileo/GPS PPP approach maintains the ability to achieve sub-nanosecond accuracy (with an average STD value of 0.48 ns) across all time links, even under kinematic conditions. This performance is on par with the static PPP time transfer, demonstrating the robustness and reliability of the Galileo/GPS PPP integration in maintaining high-precision time synchronization.

To delve deeper into the analysis of the frequency stability of the PPP time transfer based on the Galileo HAS service under kinematic conditions, Figure 16 illustrates the MDEV for various time links across different system configurations. Concurrently, Table 6 presents the frequency stability for the same links under the same system combinations, measured over intervals of 960 s and 15,360 s.

Upon a comprehensive review, it is observed that, with the exception of the TWTF-PTBB link, the GPS PPP exhibits a higher MDEV in kinematic mode compared to Galileo PPP. Moreover, the integrated Galileo/GPS PPP demonstrates a markedly superior frequency stability over any single-system PPP time transfer in kinematic conditions. At the 960 s frequency stability mark, the average Median Deviation (MDEV) for the distinct time links in the GPS, Galileo, and combined Galileo/GPS PPP solutions are 5.29 × 10^−13^, 3.34 × 10^−13^, and 1.60 × 10^−13^, respectively. Extending the observation to a 15,360 s frequency stability duration, the corresponding averages are 1.49 × 10^−13^, 1.02 × 10^−13^, and 4.06 × 10^−14^, respectively. In comparison with the static PPP time transfer, the dynamic time transfer exhibits considerably poorer frequency stability. Consequently, there remain a multitude of challenges to be addressed and investigated in the pursuit of high-precision timing in dynamic environments.

## 5. Summary

The Galileo HAS delivers high-precision Positioning, Navigation, and Timing (PNT) capabilities on a global scale. As the advent of intelligent and autonomous devices gains momentum, the study of PPP time transfer has become a focal point of interest in the kinematic model. This article delves into the characteristics of Galileo/GPS PPP time transfer facilitated by the Galileo HAS service in the kinematic model and presents several key findings.

Initially, employing the single GPS PPP approach, which relies on the Galileo HAS service, is advised against for one-way timekeeping. Under static conditions, the PPP time transfer for GPS, Galileo, and the combined Galileo/GPS systems all achieve sub-nanosecond precision. The Galileo-only PPP time transfer outperforms the GPS-only PPP, with the Galileo/GPS PPP time transfer demonstrating the most superior performance. Regarding frequency stability, GPS, Galileo, and the Galileo/GPS PPP systems can attain levels of approximately (5.80 × 10^−14^, 4.82 × 10^−14^, 4.07 × 10^−14^) at 960 s, respectively, and approximately (1.19 × 10^−14^, 1.13 × 10^−14^, 8.69 × 10^−15^) at the extended stability duration of 15,360 s.

In the realm of PPP time transfer facilitated by the Galileo HAS service in the kinematic model, the GPS-only and Galileo-only PPP time transfer methods often encounter convergence issues due to a limited number of satellites, which can significantly impair the accuracy of time transfer in the kinematic model, exceeding the threshold of 1 nanosecond. Nonetheless, when instances of re-convergence are rare, these single-system PPP methods can still achieve sub-nanosecond synchronization. Happily, the Galileo/GPS combined PPP time transfer maintains a commendable level of precision, approximately at 0.48 nanoseconds, in the kinematic. Consequently, it is advisable to employ a multi-system fusion approach for PPP time transfer endeavors in the kinematic model. Regarding frequency stability, PPP time transfer exhibits a markedly lower accuracy compared to static PPP time transfer in the kinematic model. For short-term stability over 960 s, the GPS, Galileo, and Galileo/GPS PPP systems achieve approximate stability levels of (5.29 × 10^−13^, 3.34 × 10^−13^, and 1.60 × 10^−13^), respectively. When considering long-term stability over 15,360 s, the respective figures are (1.49 × 10^−13^, 1.02 × 10^−13^, and 4.06 × 10^−14^).

## Figures and Tables

**Figure 1 sensors-25-03243-f001:**
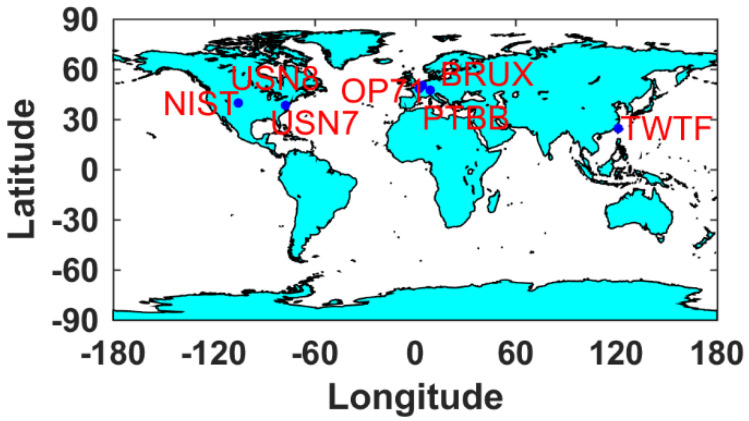
The selected GNSS station distribution map.

**Figure 2 sensors-25-03243-f002:**
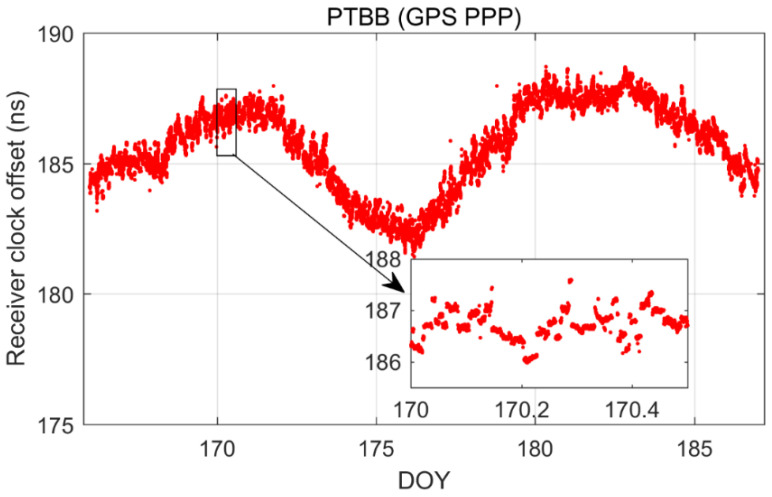
The receiver clock offset from GPS PPP solutions in the static model for PTBB station.

**Figure 3 sensors-25-03243-f003:**
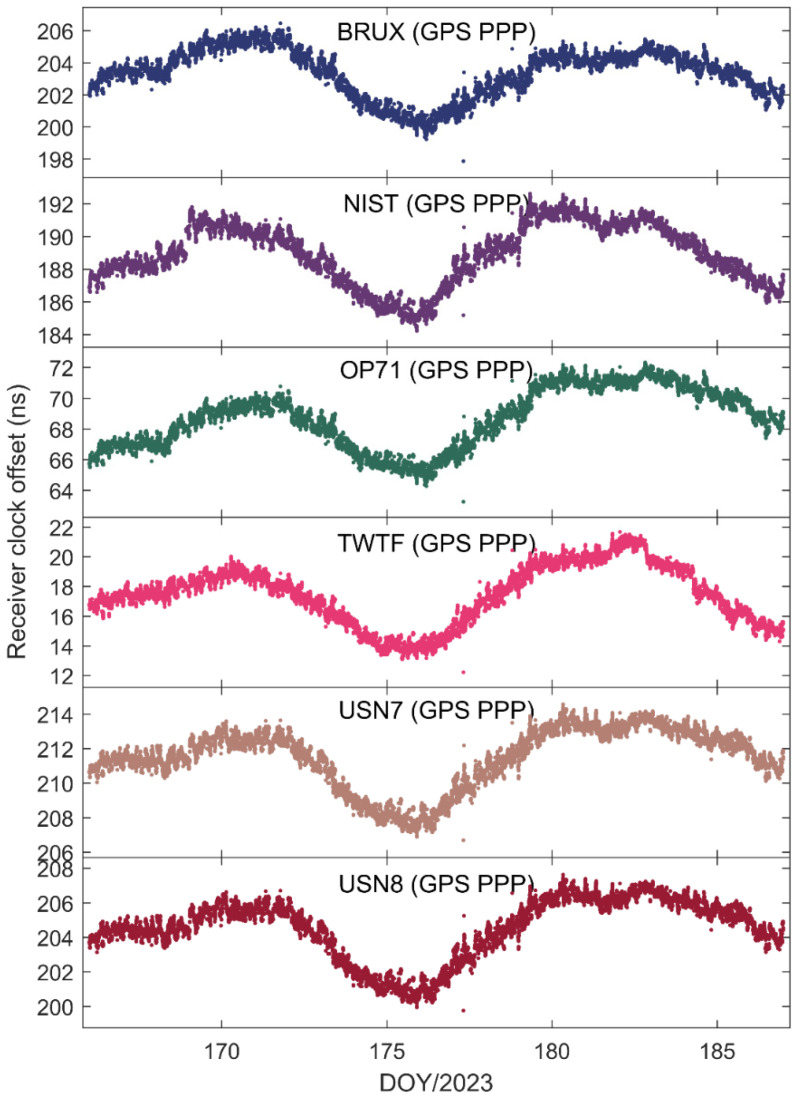
The receiver clock offset from GPS PPP solutions in the static model for six stations.

**Figure 4 sensors-25-03243-f004:**
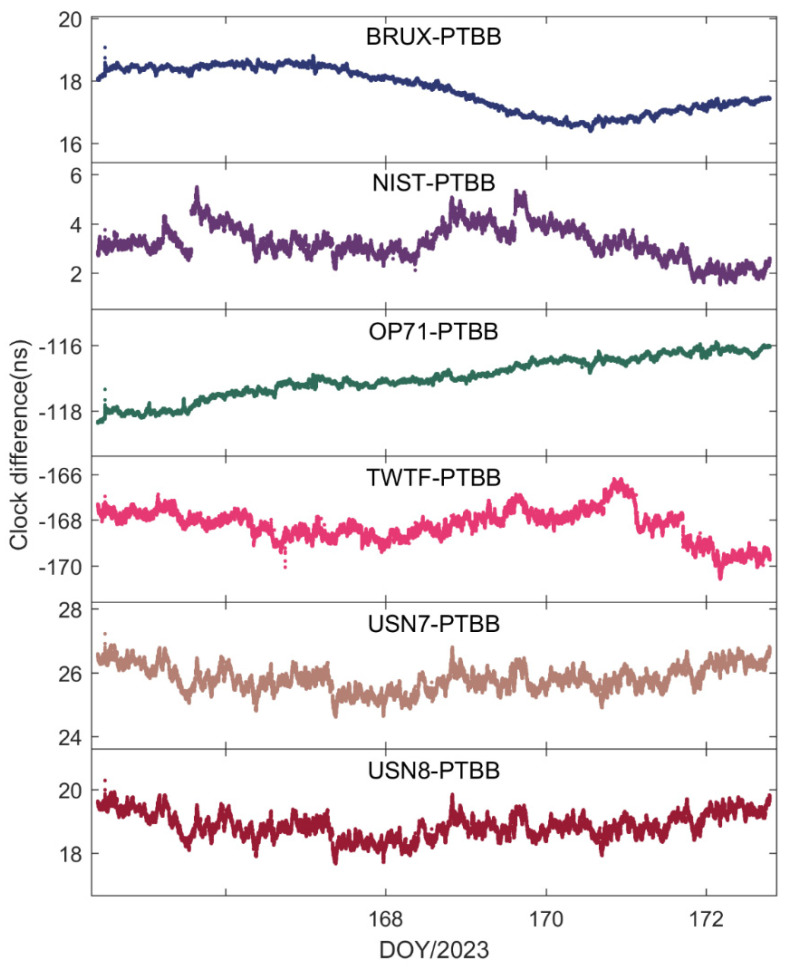
GPS PPP time transfer solutions in the static model.

**Figure 5 sensors-25-03243-f005:**
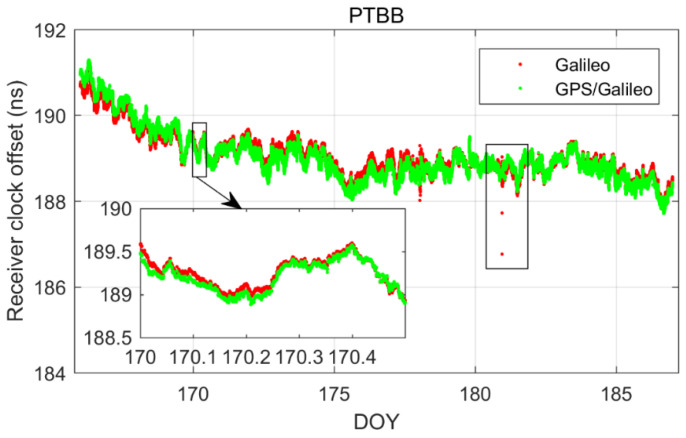
Galileo/GPS or Galileo PPP in the static model for PTBB station.

**Figure 6 sensors-25-03243-f006:**
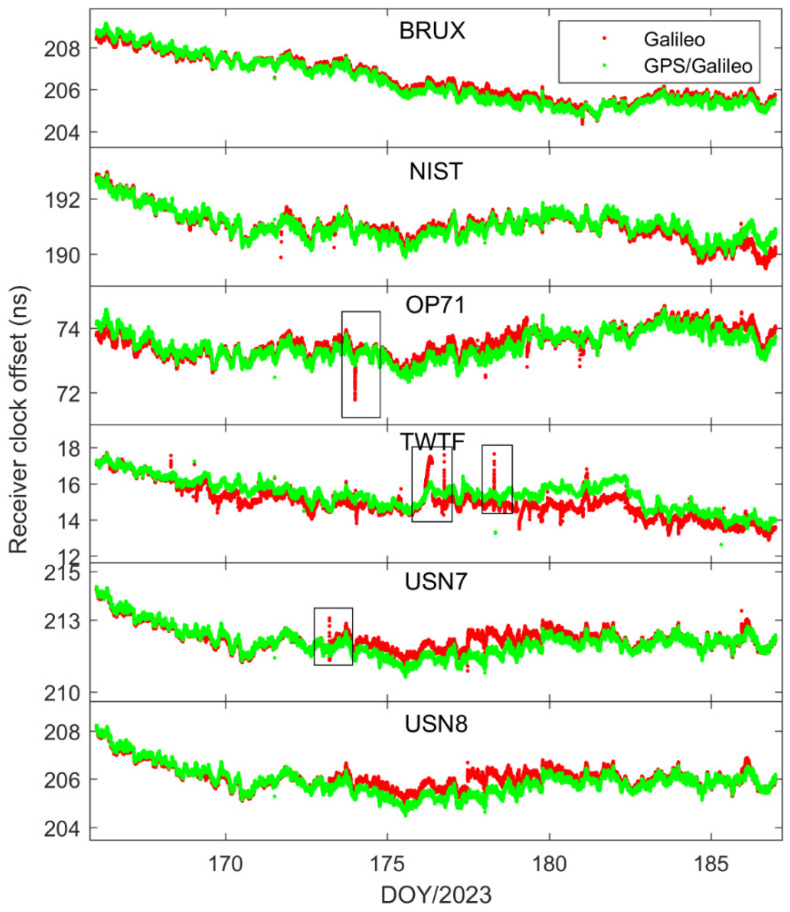
Galileo/GPS or Galileo PPP in the static model for six stations.

**Figure 7 sensors-25-03243-f007:**
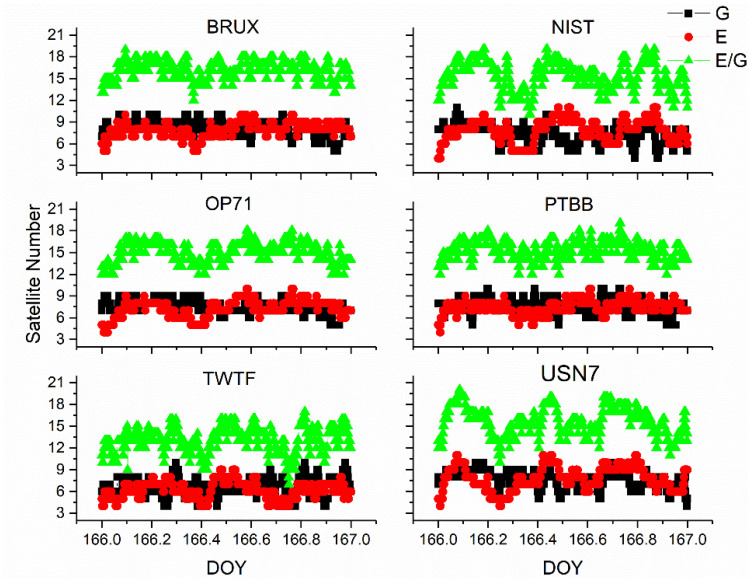
The satellite number for GPS (G), Galileo (E), or Galileo/GPS (E/G) PPP time transfer.

**Figure 8 sensors-25-03243-f008:**
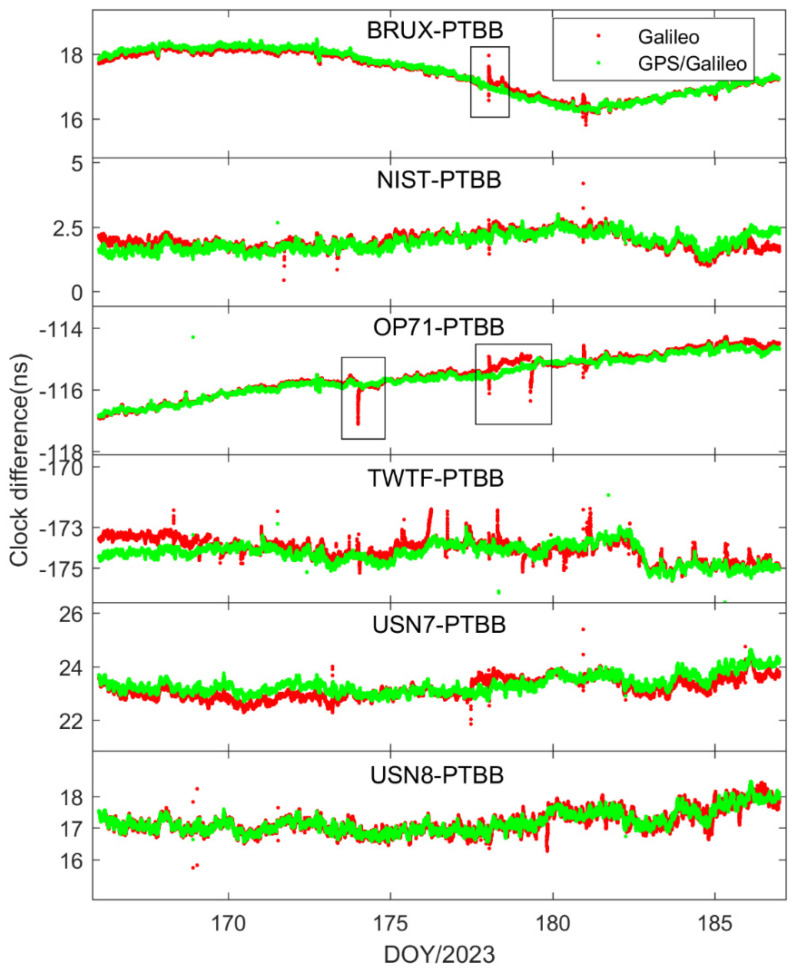
Galileo/GPS or Galileo PPP time transfer solutions in the static model.

**Figure 9 sensors-25-03243-f009:**
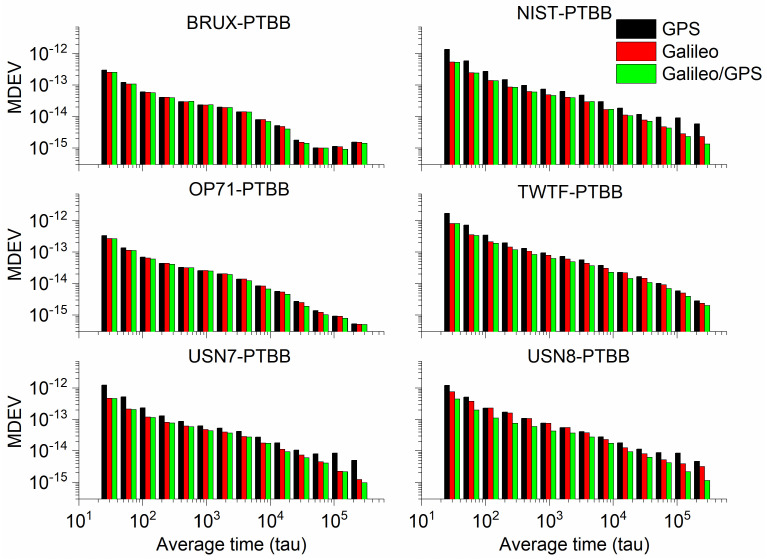
The MDEV of GPS, Galileo, or Galileo/GPS PPP time transfer in the static model.

**Figure 10 sensors-25-03243-f010:**
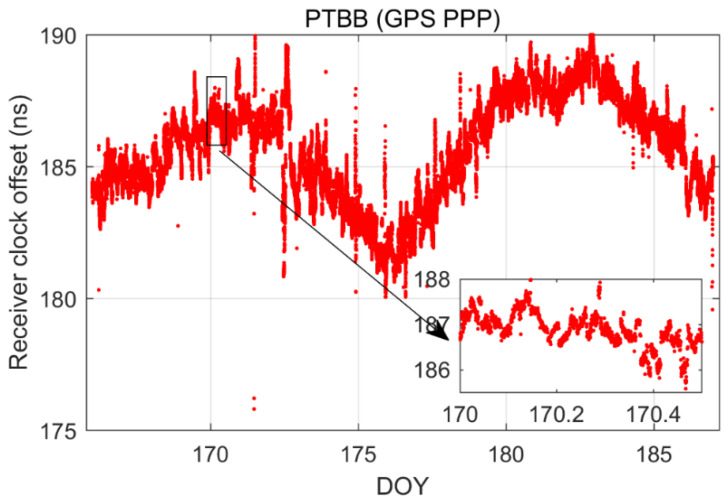
The receiver clock offset from GPS PPP in the kinematic model for the PTBB station.

**Figure 11 sensors-25-03243-f011:**
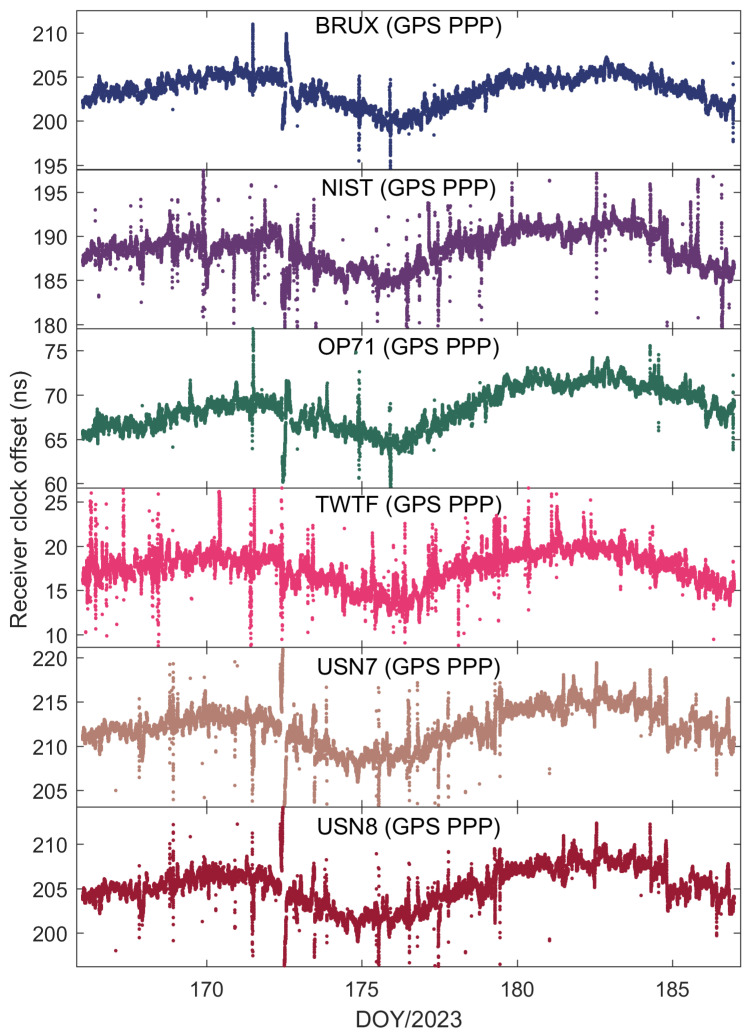
The receiver clock offset from GPS PPP in the kinematic model for six stations.

**Figure 12 sensors-25-03243-f012:**
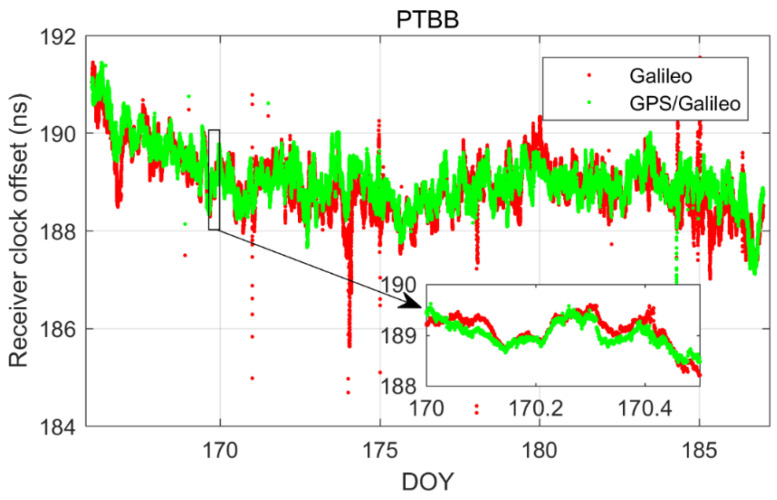
Galileo\GPS PPP in the kinematic model for PTBB station.

**Figure 13 sensors-25-03243-f013:**
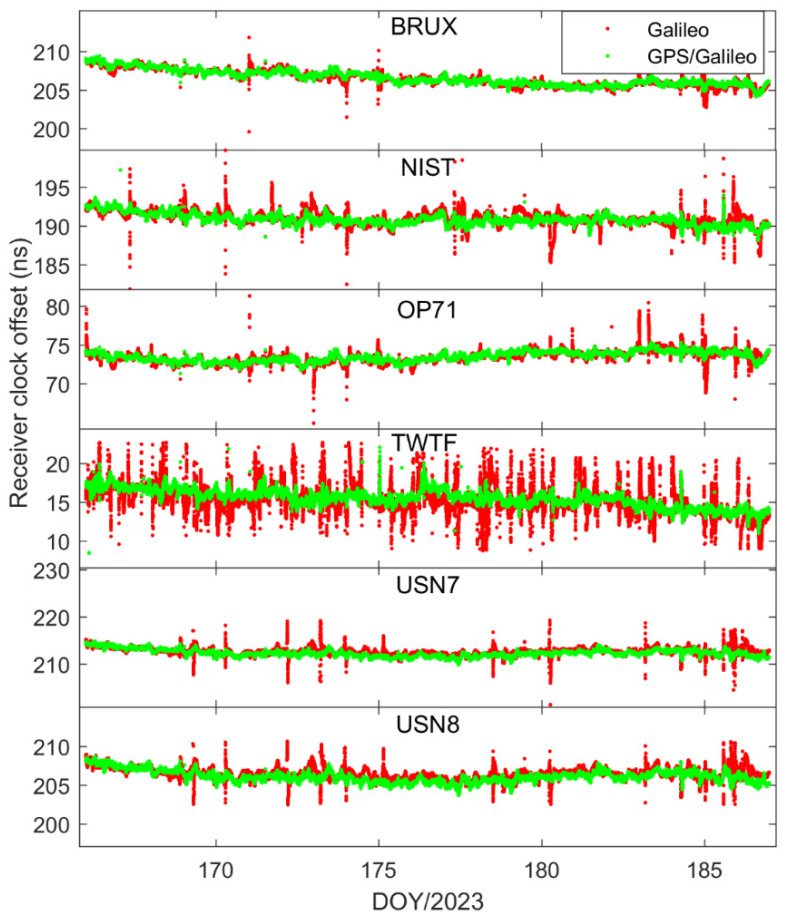
Galileo/GPS or Galileo PPP in the kinematic model for six stations.

**Figure 14 sensors-25-03243-f014:**
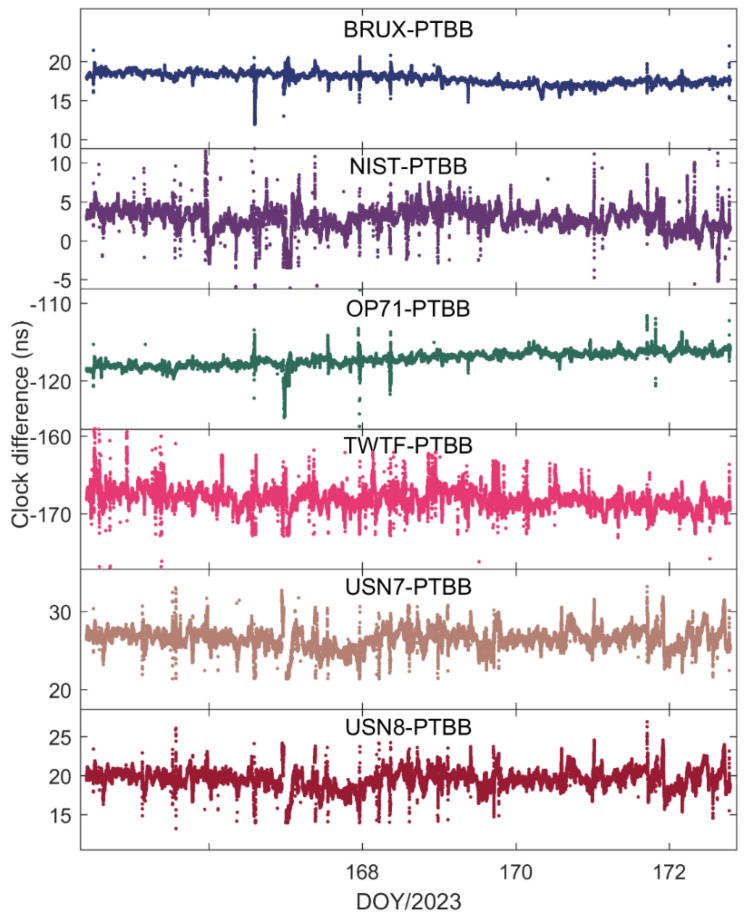
GPS PPP time transfer in the kinematic model for six time links.

**Figure 15 sensors-25-03243-f015:**
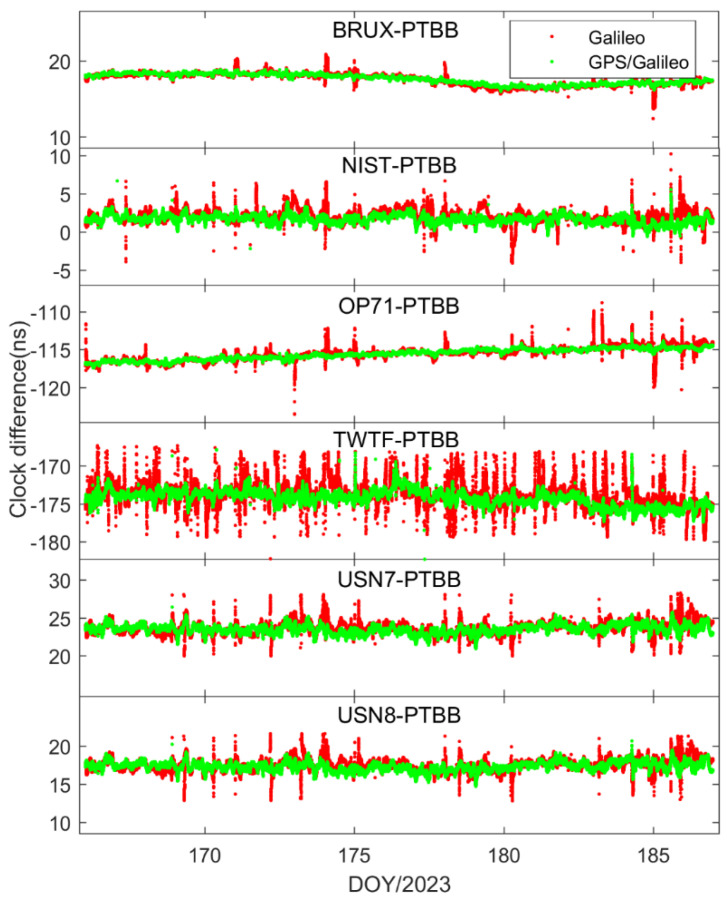
Galileo/GPS or Galileo PPP time transfer solutions in the kinematic model for six time links.

**Figure 16 sensors-25-03243-f016:**
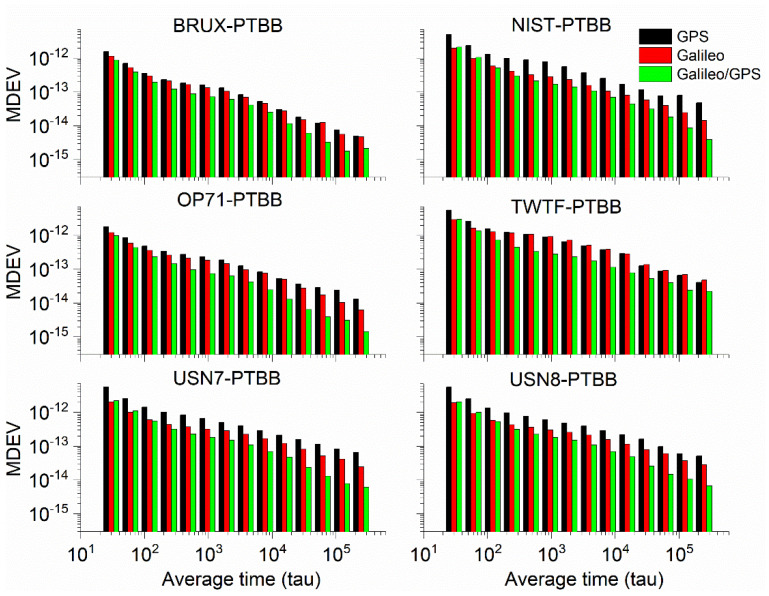
The MDEV of GPS, Galileo, or Galileo/GPS PPP time transfer in the kinematic model.

**Table 1 sensors-25-03243-t001:** The information on the selected stations.

Station	Receiver	Antenna	Clock
PTBB	SEPT POLARX5TR	LEIAR25.R4	UTC (PTB)
BRUX	SEPT POLARX5	TRM59800.00	UTC (ROB)
OP71	SEPT POLAX5TR	LEIAR25.R4	UTC (OP)
NIST	SEPT POLARX5TR	NOV750.R4	UTC (NIST)
TWTF	SEPT POLARX4TR	ASH701945C_M	UTC (TL)
USN7	SEPT POLARX5TR	TPSCR.G5	H-MASER
USN8	SEPT POLARX5TR	TPSCR.G5	H-MASER

**Table 2 sensors-25-03243-t002:** Overview of the PPP approach utilized in this study.

Estimator	Kalman Filter
Signals	Galileo: E1/E5a GPS: L1/L2
PCV (Phase Center Variation) and PCO (Phase Center Ofset)	Corrected by “igs20.atx”
Receiver clock offset	Estimated with a white noise model (10^4^ m^2^)
Precise products	Broadcast ephemeris + HAS correction
	IGS final products
Tropospheric delay	ZHD (Zenith Hydrostatic Delay): corrected [33]
	ZWD (Zenith Wet Delay): estimated with a random walk noise model (3 × 10^−8^ m^2^/s)
Tidal displacement	Corrected [34]
Phase ambiguities	Estimate as constant at each arc; when cycle-slip happened, estimated as white noise model (10^4^ m^2^)
Receiver coordinates	Static model: Estimate as constant
	Kinematic model: Estimate as white noise (10^4^ m^2^)

**Table 3 sensors-25-03243-t003:** The STD values of GPS, Galileo, or Galileo/GPS PPP time transfer in the static model (unit: ns).

Time-Links	GPS	Galileo	Galileo/GPS
BRUX-PTBB	0.26	0.21	0.22
NIST-PTBB	0.41	0.30	0.34
OP71-PTBB	0.23	0.19	0.18
TWTF-PTBB	0.62	0.41	0.40
USN7-PTBB	0.47	0.39	0.35
USN8-PTBB	0.45	0.38	0.34

**Table 4 sensors-25-03243-t004:** The MDEV of GPS (G), Galileo (E), and Galileo/GPS (E/G) PPP in the static model at 960 s and 15,360 s for six time-links (unit: ns).

Time-Links	960 s			15,360 s		
	G	E	E/G	G	E	E/G
BRUX-PTBB	2.37 × 10^−14^	2.38 × 10^−14^	2.35 × 10^−14^	4.82 × 10^−15^	4.20 × 10^−15^	4.04 × 10^−15^
NIST-PTBB	7.44 × 10^−14^	4.92 × 10^−14^	4.64 × 10^−14^	1.61 × 10^−14^	1.56 × 10^−14^	1.06 × 10^−14^
OP71-PTBB	2.59 × 10^−14^	2.58 × 10^−14^	2.50 × 10^−14^	5.40 × 10^−15^	5.39 × 10^−15^	4.53 × 10^−15^
TWTF-PTBB	9.49 × 10^−14^	7.80 × 10^−14^	6.24 × 10^−14^	2.18 × 10^−14^	2.08 × 10^−14^	1.43 × 10^−14^
USN7-PTBB	6.26 × 10^−14^	4.75 × 10^−14^	4.38 × 10^−14^	1.11 × 10^−14^	1.10 × 10^−14^	9.31 × 10^−15^
USN8-PTBB	6.67 × 10^−14^	6.51 × 10^−14^	4.34 × 10^−14^	1.24 × 10^−14^	1.10 × 10^−14^	9.36 × 10^−15^

**Table 5 sensors-25-03243-t005:** The STD values of GPS, Galileo, or Galileo/GPS PPP time transfer in the kinematic model.

Time-Links	GPS	Galileo	Galileo/GPS
BRUX-PTBB	0.37	0.33	0.23
NIST-PTBB	1.26	1.05	0.58
OP71-PTBB	0.35	0.35	0.19
TWTF-PTBB	1.09	1.57	0.76
USN7-PTBB	1.55	1.08	0.58
USN8-PTBB	1.46	1.10	0.58

**Table 6 sensors-25-03243-t006:** The MDEV of GPS (G), Galileo (E), and Galileo/GPS (E/G) PPP in the static model at 960 s and 15,360 s for six time links.

Time-Links	960 s			15,360 s		
	G	E	E/G	G	E	E/G
BRUX-PTBB	1.62 × 10^−13^	1.34 × 10^−13^	7.22 × 10^−14^	3.06 × 10^−14^	2.80 × 10^−14^	1.16 × 10^−14^
NIST-PTBB	7.89 × 10^−13^	2.84 × 10^−13^	1.72 × 10^−13^	1.70 × 10^−13^	8.05 × 10^−14^	4.45 × 10^−14^
OP71-PTBB	2.37 × 10^−13^	1.82 × 10^−13^	7.49 × 10^−14^	5.41 × 10^−14^	5.04 × 10^−14^	1.30 × 10^−14^
TWTF-PTBB	7.26 × 10^−13^	7.83 × 10^−13^	2.82 × 10^−13^	2.07 × 10^−13^	2.18 × 10^−13^	7.84 × 10^−14^
USN7-PTBB	6.55 × 10^−13^	3.21 × 10^−13^	1.83 × 10^−13^	2.15 × 10^−13^	1.22 × 10^−13^	4.78 × 10^−14^
USN8-PTBB	6.09 × 10^−13^	3.04 × 10^−13^	1.79 × 10^−13^	2.18 × 10^−13^	1.16 × 10^−13^	4.83 × 10^−14^

## Data Availability

The datasets analyzed in this study are managed by IGS.

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
