# Peer review of "An Investigation of Real-Time Galileo/GPS Integrated Precise Kinematic Time Transfer Based on Galileo HAS Service"

_sensors, 2025, doi:10.3390/s25103243_

Round 1
Reviewer 1 Report
Comments and Suggestions for Authors
See detailed comments in the file

Reviewer 2 Report
Comments and Suggestions for Authors
- The references [30] and [31] are cited in text but are not defined and appeared on the separated page (rows 575-576)
- Coefficients alpha and beta in the equation (3) are not named as "frequency factors"
- The antenna parameters PCV and PCO at Table 2 should be written without abbreviations too. In general a section "Abbreviations" should be added at the paper end. For example this includes the abbreviations ZWD, ZHD, etc. in Table 2.
- The estimator at Table 2 is defined as Kalman filter, but no more details are given for this filter
- The text and the figures should be rearranged to remove the big empty spaces in the pages
- The equation (6) is not mentioned in [27]. Also the parameter delta in the same equation is not defined (for all RAC directions)
- The english language may be improved especially the definite article
Reviewer 3 Report
Comments and Suggestions for Authors
The Galileo HAS service can achieve high-precision services in network-free environments. With the rapid development of smart cities and intelligent transportation, the demand for high-precision timing from dynamic carriers is increasing. This manuscript pioneers the study of PPP-based dynamic time transfer based on the Galileo HAS service, which is a valuable research effort and highly recommended for publication in this journal. However, there are still some minor issues with the manuscript that require revision, and it is recommended to publish it after making the necessary corrections.
- In Equations (1) and (2), the symbols on the left-hand side are currently described as observations. However, considering that the geometric distance term on the right-hand side is expressed in an incremental form, it would be more appropriate to describe them as O-C.
- In Table 2, the signals B1I/B3I are incorrect, as these actually belong to the BDS. This should be revised accordingly.
- In Tables 3 and 5, the units for STD are not specified.
- In Fig. 11, one data point from NIST is marked as selected. Please replace it with an unmarked version of the figure.
Round 2
Reviewer 1 Report
Comments and Suggestions for Authors
The authors properly addressed all my comments.
Author Response
Thanks for your suggestions.